# The impact of COVID-19 vaccination campaigns accounting for antibody-dependent enhancement

**Nessma Adil Mahmoud Yousif**[1,2], **Henri Christian Junior Tsoungui Obama**[1,2], **Yvan Jordan Ngucho Mbeutchou**[2], **Sandy Frank Kwamou Ngaha**[1,2], **Loyce Kayanula**[1,2], **George Kamanga**[1,2], **Toheeb Babatunde Ibrahim**[1,2], **Patience Bwanu Iliya**[1,2], **Sulyman Iyanda**[1,2], **Looli Alawam Nemer**[1,2], **Kristina Barbara Helle**[1], **Miranda Ijang Teboh-Ewungkem**[3], **Kristan Alexander Schneider**[1]*

**1** Department of Applied Computer- and Biosciences, University of Applied Sciences Mittweida, Mittweida, Germany, **2** African Institute for Mathematical Sciences Cameroon, Limbe, Cameroon, **3** Department of Mathematics, Lehigh University, Bethlehem, PA, United states of America

☯ These authors contributed equally to this work.
* kristan.schneider@hs-mittweida.de

## Abstract

### Background

COVID-19 vaccines are approved, vaccination campaigns are launched, and worldwide return to normality seems within close reach. Nevertheless, concerns about the safety of COVID-19 vaccines arose, due to their fast emergency approval. In fact, the problem of antibody-dependent enhancement was raised in the context of COVID-19 vaccines.

### Methods and findings

We introduce a complex extension of the model underlying the pandemic preparedness tool CovidSim 1.1 (http://covidsim.eu/) to optimize vaccination strategies with regard to the onset of campaigns, vaccination coverage, vaccination schedules, vaccination rates, and efficiency of vaccines. Vaccines are not assumed to immunize perfectly. Some individuals fail to immunize, some reach only partial immunity, and—importantly—some develop antibody-dependent enhancement, which increases the likelihood of developing symptomatic and severe episodes (associated with higher case fatality) upon infection. Only a fraction of the population will be vaccinated, reflecting vaccination hesitancy or contraindications. The model is intended to facilitate decision making by exploring ranges of parameters rather than to be fitted by empirical data. We parameterized the model to reflect the situation in Germany and predict increasing incidence (and prevalence) in early 2021 followed by a decline by summer. Assuming contact reductions (curfews, social distancing, etc.) to be lifted in summer, disease incidence will peak again. Fast vaccine deployment contributes to reduce disease incidence in the first quarter of 2021, and delay the epidemic outbreak after the summer season. Higher vaccination coverage results in a delayed and reduced epidemic peak. A coverage of 75%–80% is necessary to prevent an epidemic peak without further drastic contact reductions.

**Data Availability Statement:** All data used is simulated data that can be reproduced with the code available on GitHub at https://github.com/Maths-against-Malaria/COVID19_ADE_Model.git.

**Funding:** This study was supported in the form of funding by the German Academic Exchange (Project-ID 57417782) awarded to KAS, Sächsisches Staatsministerium für Wissenschaft und Kunst (Project number 100257255) awarded to KAS, the Federal Ministry of Education and Research (BMBF) and the DLR (Project-ID 01DQ20002) awarded to KAS, the Europäischer Sozialfond (ESF) Young Investigator Group "Agile Publika" funded by ESF, SMWK, SAB (SAB Project 100310497) awarded to KAS and KBH, and the Deutsche Forschungsgemeinschaft (DFG), Project-ID 656983 awarded to KAS.

**Competing interests:** The authors have declared that no competing interests exist.

## Conclusions

With the vaccine becoming available, compliance with contact reductions is likely to fade. To prevent further economic damage from COVID-19, high levels of immunization need to be reached before next year's flu season, and vaccination strategies and disease management need to be flexibly adjusted. The predictive model can serve as a refined decision support tool for COVID-19 management.

## Introduction

Interventions such as curfews, lockdowns, cancellations of mass events, in response to the COVID-19 pandemic caused massive losses in revenue for whole economic sectors [1, 2]. They were justified as legitimate measures to delay the epidemic to increase healthcare capacities, develop effective treatments, and vaccines to immunize the population. Despite all efforts, the spread of COVID-19 since fall 2020 drives healthcare systems to their limits in Europe and North America, underlining the urge for an effective and safe vaccine.

Governments across the globe were ambitious in facilitating SARS-CoV-2 vaccine development, by large-scale programs such as *Operation Warp Speed* launched by the US Government [3]. Currently, more than 227 vaccine-development projects against SARS-CoV-2 are ongoing [4]. There are four major vaccination platforms to stimulate antibody production triggered by the SARS-CoV-2 spike protein: (i) viral vectors fused with a gene that encodes for the SARS-CoV-2 spike protein; (ii) inactivated SARS-CoV-2 variants; (iii) protein subunits of SARS-CoV-2 antigens; and (iv) a rather new technique, where lipid nanoparticles encapsulate nucleoside-modified mRNA (modRNA) encoding mutated forms of the SARS-CoV-2 spike protein. The most promising candidates typically follow a 2-3 week vaccination schedule, after whose completion the protective effect is reached within 2-3 weeks [5, 6].

Russia was ambitious to release the world's first SARS-CoV-2 vaccine *Gam-COVID-Vac*. In homage named *Sputnik V*, the vaccine is based on two human adenovirus (common cold) vectors [7], costs less than 20 USD per dose, and is highly controversial due to its fast emergency use authorization (EUA) without a phase III study [8]. Nevertheless, mass vaccination started in Russia on December 5, 2020. The modified chimpanzee adenovirus vector-based candidate from AstraZeneca, *AZD1222*, is currently under phase III study, will cost only 4 USD per dose, and has a capacity of 400 million doses for Europe and 300 million doses for the USA.

China gave EUA to two vaccines that trigger an immune response by inactivated SARS-CoV-2 variants. *BBIBP-CorV* has a capacity of 1 billion doses for China in 2021 at a cost of less than 75 USD per dose and was fully approved, while *CoronaVac* costs 30 USD per dose. *Covaxin (BBV152)* is a cheap vaccine (1.36 USD per dose) currently under EUA developed by the Indian Council of Medical Research, based on inactivated SARS-CoV-2 variants.

*NVX-CoV2373* by Novavax, seeking approval in Mexico, is a vaccine that uses SARS-CoV-2 recombinant spike protein nanoparticles with adjuvants to trigger an immune response [9].

Two modRNA-based candidates are currently in phase III studies, which either seek approval or were granted EUA. *Tozinameran (BNT162b2)* by BioNTech (20 USD per dose), was approved in Canada and Europe, and received EUA in the UK and the USA. A second modRNA-based candidate, *mRNA-1273* by Moderna, is currently in phase III trials, and received EUA in Canada and the USA.

Vaccination campaigns aim for herd immunity. There is an ongoing debate on the optimal deployment of the vaccine. Some countries have ambitious deployment strategies, e.g.,

Morocco plans to immunize up to 80% of the population. Globally the trend is to deploy vaccines voluntarily and free of charge, with a general agreement to prioritize vulnerable risk groups (e.g., senior citizens, people with co-morbidities, etc.) and people of systemic importance (e.g., healthcare workers, police, public services) before making the vaccine available to the general public [10]. Incentives for getting voluntary vaccines have been proposed, e.g., recently Qantas airlines announced to make the vaccine mandatory for their passengers [11, 12].

Nevertheless, skepticism about vaccines and their potential side effects are widespread, resulting in vaccine hesitancy [13]. One of the potentially negative effects of a vaccine is the occurrence of antibody-dependent enhancement (ADE) or, more general, enhanced respiratory disease (ERD) [14, 15]. ADE is best understood in Dengue fever and was observed also in SARS-CoV and MERS-CoV both in vitro and in vivo [16]. In SARS-CoV-2, ADE occurs most likely via enhanced immune activation [17]. Here, sub-optimal antibodies form immune complexes with the virus that deposit into airway tissues and activate cytokine and complement pathways. This triggers inflammation, airway obstruction, and even acute respiratory distress syndrome [17]. By this mechanism, vaccines could potentially result in more severe symptoms upon infection with SARS-CoV-2.

Here, we employ predictive modelling to explore the outcome of vaccination strategies on the COVID-19 pandemic. We explore the influence of (i) the vaccination coverage reflecting vaccine hesitancy, the commonness of contraindications, and access to the vaccine; (ii) the vaccination rate, reflecting supply and infrastructure; and (iii) the immunizing effect on disease incidence, prevalence, and mortality. We further investigate the impact of the launch of the vaccination campaign relative to the epidemic peak. Intentionally, our model accounts for the occurrence of ADE, ERD, and other deleterious side effects of the vaccine—subsumed here as ADE. By this approach, we seek to address the following questions: What is the benefit of launching the vaccination campaign early (at full scale) to substantially reduce disease incidence? Can side effects of the vaccine result in overall higher mortality than the virus itself? How fast should the vaccine be deployed? Which vaccination coverage should be aimed at? The model is intended as a preparedness tool to facilitate decision making by exploring ranges of parameters, which are difficult to be estimated from empirical data. The model is not designed to be fitted by empirical data. Rather empirical data should be used as a plausibility check for the model's parametrization.

We use an extension of the SEIR model underlying the pandemic preparedness tool Covid-Sim 1.1 [18] to predict the outcome of vaccination campaigns. The model incorporates general contact reduction (hard and soft lockdowns, social distancing, etc.) and case isolation in a time-dependent manner. Unvaccinable persons, summarizing anti-vaxxers, individuals with contraindications, and individuals that do not have access to the vaccine, are properly addressed. Furthermore, immunization after receiving the vaccine does not occur instantaneously, reflecting vaccination schedules and immunogenicity. Importantly, our model allows individuals to be vaccinated during the infection (if it is asymptomatic and undetected) and to be infected before the outcome of the vaccination manifests. As an example, we use model parameters that reflect the situation in the Federal Republic of Germany. In the main text, we present only a verbal description of the model and refer to S1 Appendix for a concise formal description, dedicated to readers interested in the technical aspects of the model.

## Methods

We model the occurrence of ADE during vaccination campaigns in the ongoing COVID-19 pandemic by an extended SEIR model. More precisely, we generalize the compartmental

model underlying the pandemic preparedness tool CovidSim (cf. [18]), which is formulated as a system of ordinary differential equations. Modelling ADE requires a substantial extension of the original model. We describe it verbally in a simplified form here and refer readers interested in a concise description to S1 Appendix.

## Course of the disease

We divide a population of size $N$ into susceptible, infected, and recovered individuals, each of which is subdivided into numerous compartments. During the course of the infection (Fig 1), individuals pass through the (i) latent phase (no symptoms, not yet infectious); (ii) prodromal phase (no symptoms, infectious but not yet to the fullest extent); (iii) fully infectious phase (symptoms might start, infectious to the fullest extent); (iv) late infectious phase (infectiousness is reduced). The fully infectious and late infectious phases can be asymptomatic or symptomatic with mild or severe symptoms. At the beginning of the fully infectious phase a fraction of infections becomes symptomatic (mild or severe symptoms), whereas the remaining fraction remains asymptomatic. At the end of the late infectious phase individuals either recover and obtain full immunity or they die. Only symptomatic infections can result in death, asymptomatic infections are not lethal (except individuals get vaccinated during the infection and this results in complications).

## Susceptible individuals

Susceptible individuals are sub-divided (see Fig 2) into those that are: (i) not vaccinable because they do not have access, refuse to be vaccinated, or cannot be vaccinated for medical reasons (e.g., allergies, or the vaccine is not approved for them); (ii) waiting to be vaccinated; (iii) already vaccinated, but the outcome of the vaccination is still pending; (iv) already vaccinated, but only partially immunized; (v) vaccinated, but the vaccination completely failed to immunize; (vi) vaccinated, but the vaccine caused ADE. Vaccinable susceptibles have to wait before being vaccinated. The outcome of the vaccination does not occur immediately but after some waiting time (pending outcome). Vaccination either results in (i) full immunity, (ii) partial immunity, (iii) no immunity (the vaccine had no effect), or (iv) ADE (the vaccine had a deleterious effect). Partial immunity gives some protection from infection and manifests in a lower likelihood to develop symptoms upon infection. ADE also protects partially from infections, but upon infection increases the likelihood of severe disease (and death).

## Effect of vaccination

Individuals are vaccinated only once (one vaccination cycle, subsuming all necessary doses). The waiting time during which immunization is pending reflects the waiting time for the vaccination cycle to be completed (it can be one or several doses). In the model we need to distinguish between infections of individuals that are (i) partially immunized, (ii) not immunized or unvaccinable, (iii) developed ADE, and (iv) vaccinable, but still waiting to be vaccinated. Vaccinable individuals (those that wait for vaccination) can also be vaccinated when they are already infected (i.e., during the latent, and prodromal phases, and during the fully infectious and late infectious phases if the infections remain asymptomatic and undetected; see Fig 3). After some waiting period (during which the effect of the vaccination is pending) these individuals are either successfully, partially, or not immunized, or they might be affected by ADE. The likelihood of these outcomes depends on the phase of the infection. Particular consideration is given to individuals that are vaccinated during the fully infectious or late infectious phase. Namely, when the disease has already progressed, the effect of the vaccine potentially changes.

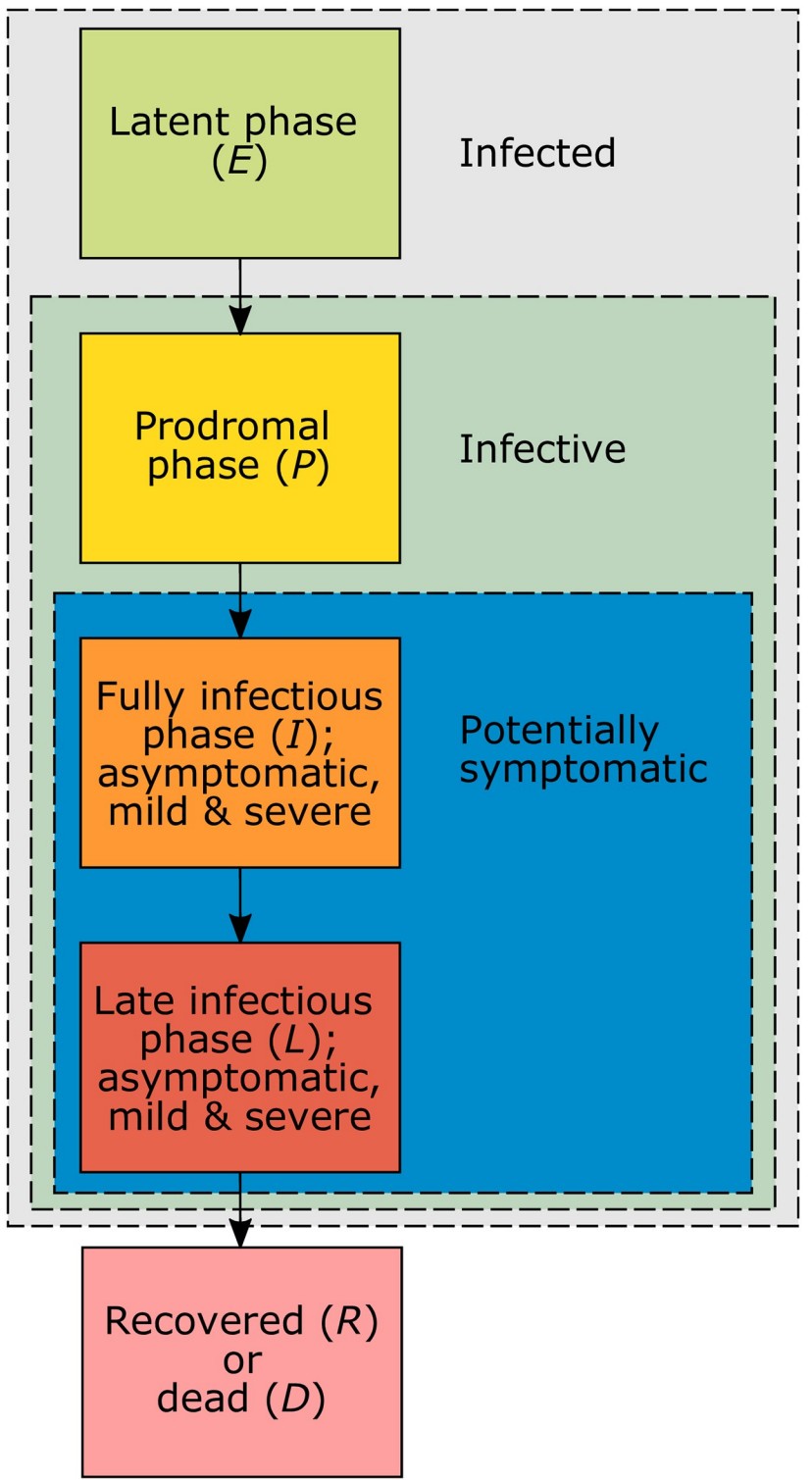

**Fig 1. Phases of the infection.** Schematic representation of the disease phases and at which stages individuals become infective and potentially symptomatic.

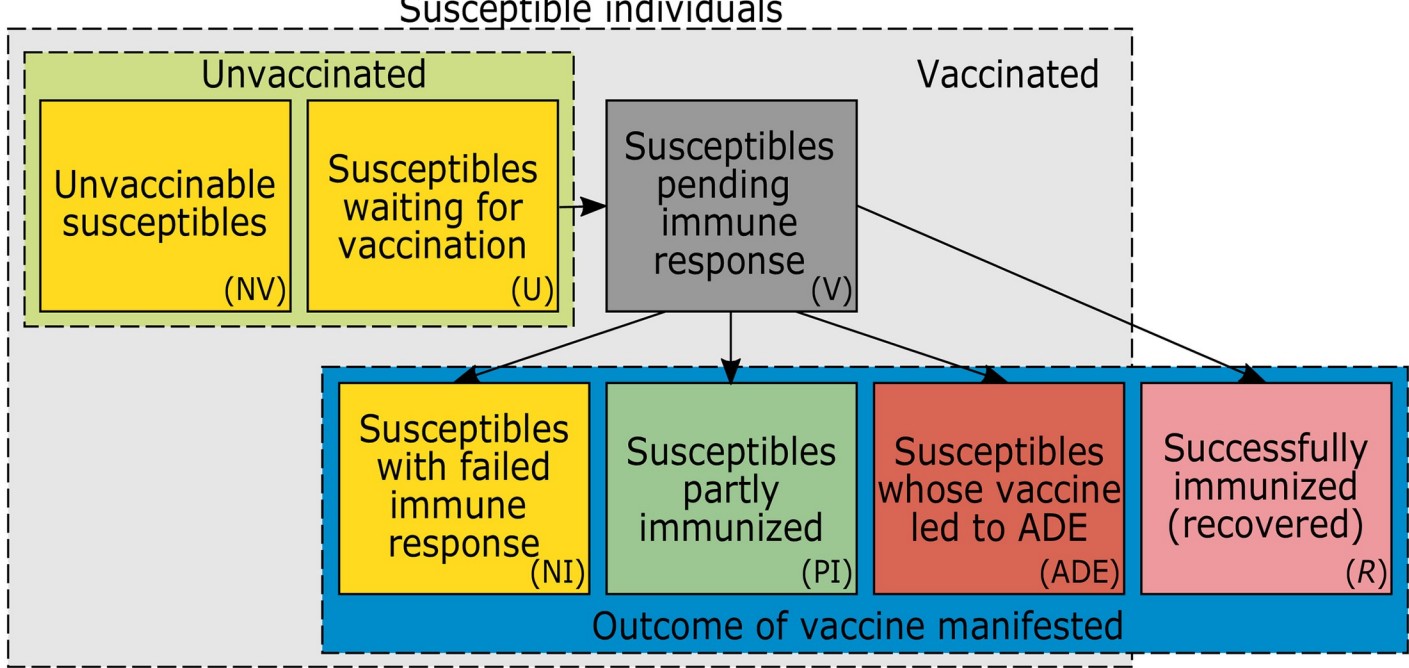

**Fig 2. Groups of susceptible and immunized individuals.** Schematic representation of different compartments of susceptible individuals. Unvaccinated susceptibles are either unvaccinable (NV) or wait to be vaccinated (U). Vaccinated susceptibles (V) remain susceptible immediately after vaccination. After the vaccination outcome manifested, they remain susceptible at different levels if immunization failed (NI), only partial immunization was achieved (PI), or they developed ADE. Individuals that are successfully immunized are no longer susceptible (R). Arrows show how individuals move between groups. All susceptibles can be infected (not indicated).

### Case isolation

A fraction of symptomatic infections seeks medical help and will be isolated in quarantine wards, until the wards are full, in which case they are sent in home isolation. Quarantine wards guarantee perfect isolation, whereas home isolation does not eliminate all infectious contacts with the isolated individuals.

### General contact reduction

During designated time intervals general contact reduction (curfews, social distancing, cancellation of mass events, etc.) is sustained. These phases reduce the number of contacts between individuals in the population. These contact reductions are time dependent. For the simulations here we assume an initial "hard lockdown" followed by a phase of relief, a "soft lockdown", a second "hard lockdown", and a final relief phase before contact reductions are lifted.

### Contact rate

Susceptible individuals get infected by random contacts with infected individuals that can transmit the disease and are not isolated. The contact rates are mediated by general contact reduction. The basic reproductive number $R_0$ is assumed to fluctuate seasonally.

### Model implementation

The model as described in detail in S1 Appendix was implemented in Python 3.8. We used a 4th order Runge-Kutta method using the function *solve ivp* as part of the library Scipy.

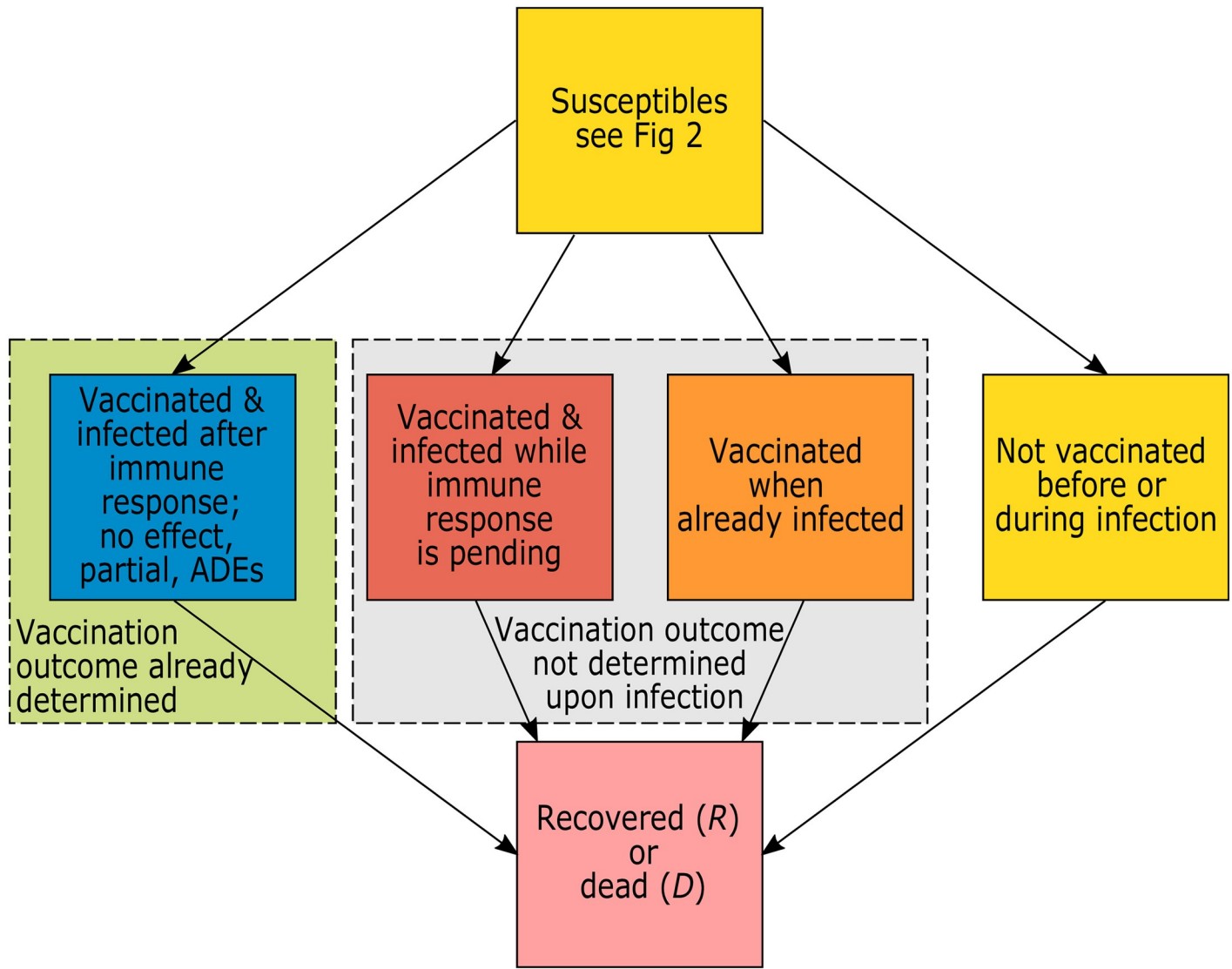

**Fig 3. Simplified model flow.** Susceptible individuals are infected and ultimately recover or die. Individuals are infected either after or before the outcome of the vaccination is determined. In the latter case the outcome of the vaccine is not determined upon infection (they might get vaccinated before or during the infection), or they are not vaccinated before or during the infection (this includes unvaccinable individuals that get infected).

Graphical output was created in R [19]. The Python code with the model implementation is available at GitHub (https://github.com/Maths-against-Malaria/COVID19_ADE_Model.git, http://doi.org/10.5281/zenodo.4659000).

## Results

Here, we study the effectiveness of different hypothetical vaccination programs/strategies to immunize the population in the ongoing COVID-19 pandemic in terms of disease incidence (or prevalence) and overall mortality. Vaccination campaigns differ in (i) their onset and vaccination rate reflecting the available infrastructure (availability of the vaccine and medical infrastructure); (ii) the vaccination coverage, reflecting the willingness of the population to get vaccinated and access to medical care; and (iii) vaccination schedules, immunogenicity, and

efficacy/effectiveness summarizing vaccine-specific properties. The ultimate goal of any vaccination campaign is to reach herd immunity, mitigate SARS-CoV-2, and "return to normality" as soon as possible. We explore how fast herd immunity is reached assuming that contact reductions cannot be sustained for too long. The approach is conservative, as we focus on the potential negative effects of the vaccine.

We report disease incidence and prevalence. Disease prevalence is defined as the sum of all infected individuals (in the latent, prodromal, fully infectious, and late infectious phases). By incidence we refer to 7-day incidence, which is the number of new cases within the last 7 days. This is defined as the integral of the force of infection over the time interval from $t - 7$ to $t$. Approximately, incidence and prevalence differ only by a multiplicative factor here and can be used synonymously. Both are reported for the readers convenience to facilitate comparison with publicly accessible data.

Regarding the results, we use model parameters reflecting the situation in the Federal Republic of Germany that so far intervened successfully in the COVID-19 pandemic. To illustrate the model's applicability to other countries, we also parameterized it to reflect the situation in the USA. The results for the USA are presented in S2 Appendix. The parameters used for Germany are listed in S1–S5 Tables. The population size of Germany was set to $N = 83$ million. We assumed the first COVID-19 cases were introduced in late February 2020 (corresponding to $t = 0$). A basic yearly average reproductive number of $\bar{R}_0 = 3.4$ was assumed, which fluctuates seasonally by 43% with a peak in late December ($t_{R_{0_{max}}} = 300$). The average durations for the latent, prodromal, fully infectious and late infectious phases were set to $D_E = 3.7$, $D_P = 1$, $D_I = 5$, $D_L = 5$ days, respectively. Individuals in the prodromal and late infectious phases were half as infectious as in the fully infectious phase ($c_P = c_L = 0.5$). If individuals developed only partial immunity or ADE, they were assumed to be half as susceptible to SARS-CoV-2 than unimmunized individuals ($p_{PI} = p_{ADE} = 0.5$). A percentage $f_{Sick} = 58\%$ of infections became symptomatic if immunity was not mediated by the vaccine. This percentage decreased for partially immunized individuals and increased for those that developed ADE (see S4 Table). These parameters were justified by CovidSim 1.1 [18] and are a combination of COVID-19 and influenza estimates. (Because the model without the vaccination is essentially equivalent to the one of CovidSim 1.1., the sensitivity of these parameters can be readily ascertained via the web simulator available at http://version-1.1.covidsim.eu/).

Regarding general contact reductions, we assumed a "hard lockdown" from early April ($t_{Dist_1} = 40$) to mid-May 2020 ($t_{Dist_2} = 82$) that reduced $p_{Dist_1} = 70\%$ of all contacts, followed by a period of relaxation until the end of October ($t_{Dist_3} = 246$) during which $p_{Dist_2} = 40\%$ of the contacts were avoided. The first "soft lockdown" from late October was sustained until the beginning of December ($t_{Dist_4} = 280$) with a contact reduction of $p_{Dist_3} = 50\%$. This was followed by a second "hard lockdown" from early December until late March ($t_{Dist_5} = 397$) and a phase of relief resulting in a $p_{Dist_4} = 68\%$ contact reduction sustained until May 2021 ($t_{Dist_6} = 450$), after which all general contact reductions are lifted reflecting the worst-case scenario, in which compliance with social distancing can no longer be sustained in the face of the vaccine becoming available. Contact reductions were deduced by assuming roughly the imposed contact restriction in Germany in schools, at work, at home, and other locations. These restrictions reduced the age-dependent contact-rate estimates available from [20], which where then averaged over all age strata, weighted by their relative sizes.

The simulation results until early March 2021 ($t = 375$) match the disease incidence in Germany. Notably, only the number of confirmed, not of actual cases is known. The true incidence (which is modelled) is obviously unobservable and might be substantially higher than the confirmed incidence (true incidence multiplied with the probability of detection).

Consequently, the incidences in Fig 4–9 exceed the number of reported cases in Germany—however, the numbers are plausible and match in the order of magnitude.

Regarding the vaccination campaigns, we use the following reference scenario: 60% of the population will get vaccinated, a vaccination rate of 1/180 (i.e., an average time of 180 days to get vaccinated), a launch of the campaign in late December 2020 ($t = 310$), and a vaccination schedule of 28 days (the vaccine becomes effective after 28 days).

## No vaccination

We consider the situation in which no vaccination campaign (VC) is launched as a reference case for comparisons. Fig 4A–4D show disease incidence, which matches the reported

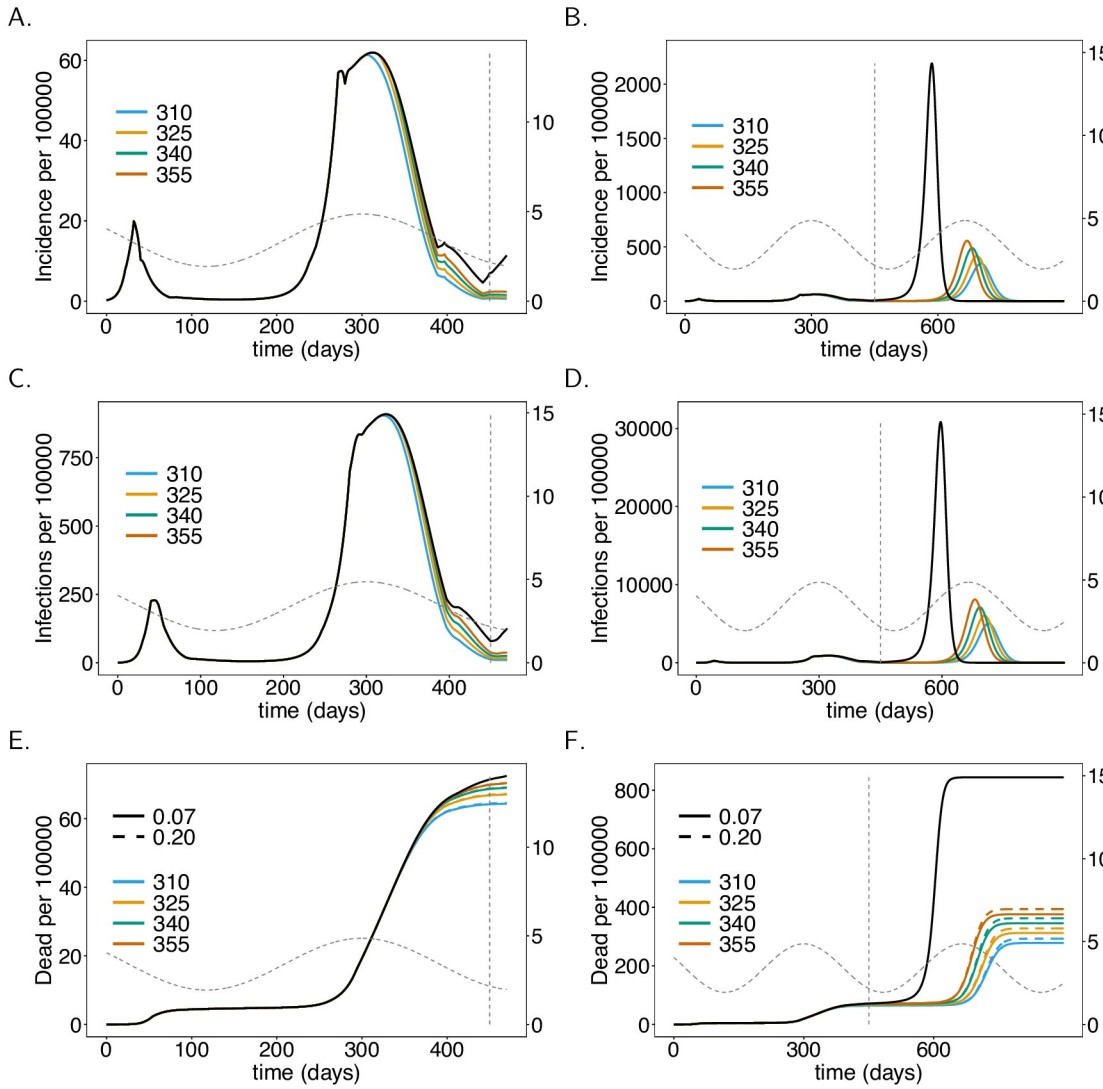

**Fig 4. Onset of the vaccination campaign.** Plots show (total) incidence (panels A-B), prevalence per 100000 individuals (panels C, D), and mortality, i.e., the cumulative deaths ($D$), per 100000 individuals (panels C, D) as a functions of time $t$ for different onsets of the vaccination campaign (colors). In panels E and F the effect of increased ADE-induced mortality is shown (dashed lines $f_{\text{Dead}}^{(\text{ADE})} = 20\%$ *vs.* solid lines $f_{\text{Dead}} = f_{\text{Dead}}^{(\text{ADE})} = 7\%$). (Note the values of $f_{\text{Dead}}^{(\text{ADE})}$ only affects mortality, not incidence.) As a baseline comparison, the black lines show incidence and mortality in the absence of the vaccine. The vertical dashed line indicates time $t = 450$ at which contact reductions are lifted. Seasonal fluctuations in $R_0$ are shown by the grey dashed lines corresponding to the y-axis on the right-hand side. Plot parameters are given in S1–S3 Tables.

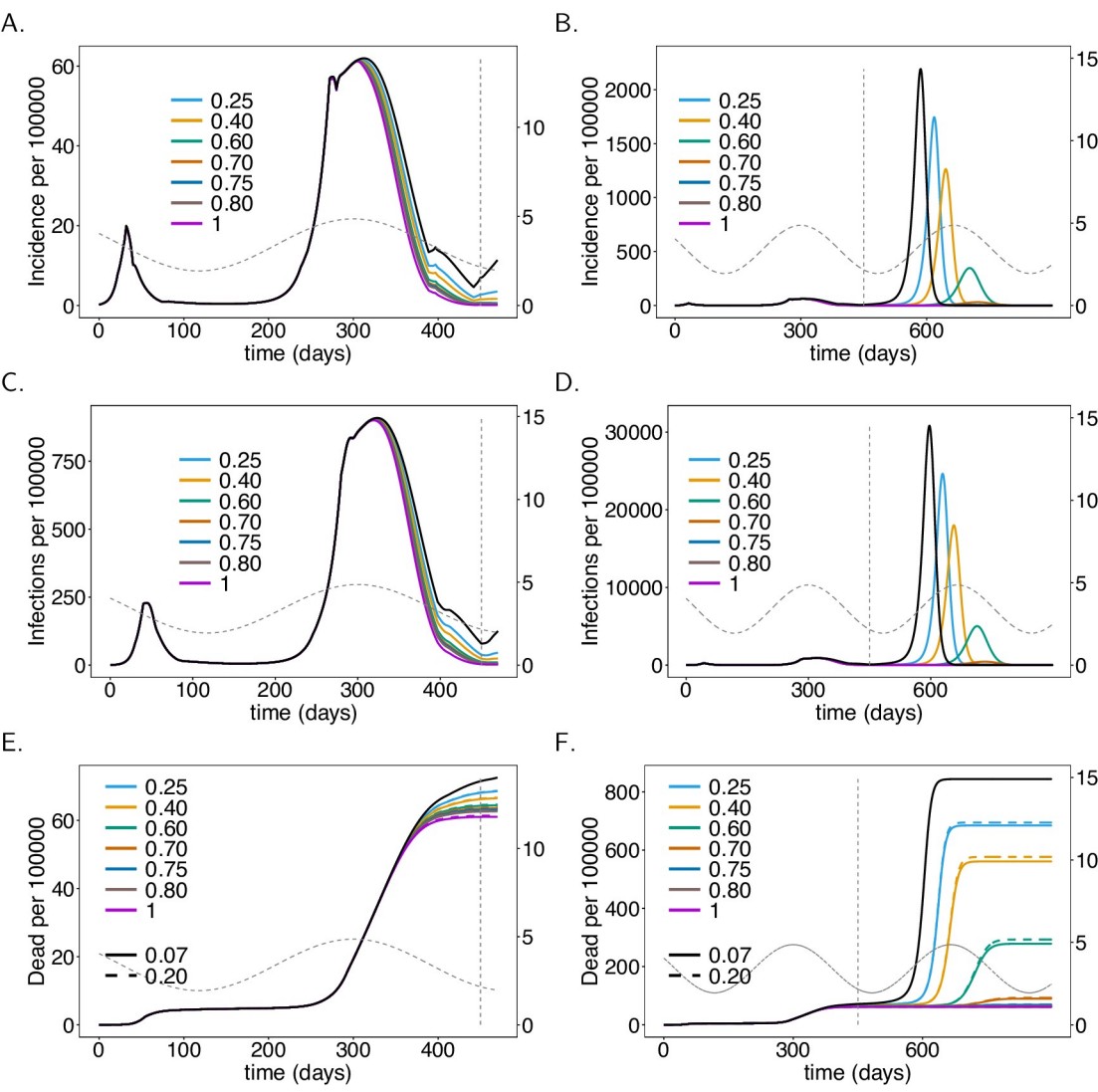

**Fig 5. Vaccination coverage.** As Fig 4, but for different vaccination coverage (colors).

numbers in Germany. The base reproductive number is at the seasonal maximum at times $t = 300$ and $t = 665$. Although $R_0$ declines after $t = 300$, the end of the second "hard lockdown" in late March ($t = 397$) leads to a moderate increase in incidence. In fact, it will start to decline later in spring until summer due to seasonal reductions in $R_0$. After contact reductions are lifted at time $t = 450$, disease incidence would increase drastically and the epidemic peak would be reached around $t = 590$ (late October 2021) with almost 40% of the population being infected at this time point. This is a hypothetical worst-case scenario.

## Onset of vaccination campaigns

The onset of the vaccination campaign (VC) has a profound effect on disease incidence (see Fig 4). The earlier the onset of the VC, the earlier and the stronger the reduction in incidence after the second "hard lockdown". Even if the VC starts at $t = 300$ incidence will increase after

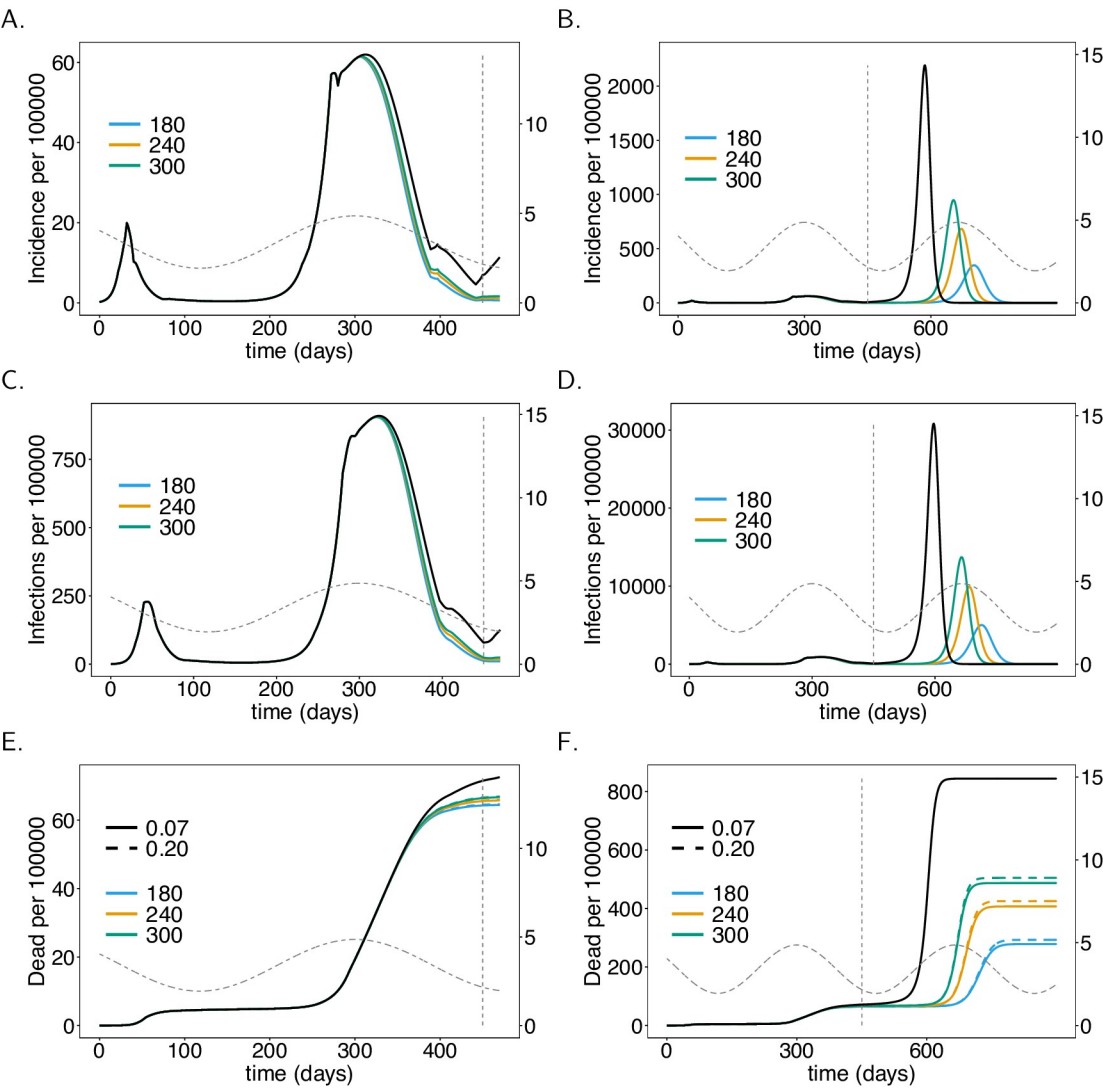

**Fig 6. Vaccination rate.** As Fig 4, but for different vaccination rates (colors), i.e., average waiting times (shown in legend) to get vaccinated.

the second "hard lockdown". However, the increase is less than half of that observed without a VC. Even launching the campaign in late February 2021 ($t = 355$) leads to a reduction in disease incidence until summer 2021 (Fig 4A and 4C). (The launch of the VC has to be interpreted as the time when it becomes fully effective).

A much stronger effect of the VC's onset manifests after contact reduction is no longer sustained. High epidemic peaks will emerge at the end of 2021 or in early 2022. Later launches of the VC result in earlier and higher epidemic peaks (Fig 4B and 4D). The differences in height of the epidemic peaks are substantial in comparison to disease incidence in 2020. In any case, the benefit of the VCs is clear. The epidemic peaks will be substantially lower compared to the situation without a VC. The earlier the vaccination campaign starts, the higher the reduction in mortality. Indeed the differences by the end of June 2021 ($t = 450$) are visible (Fig 4E). The reduction in mortality by the end of March 2022 is substantial (Fig 4F).

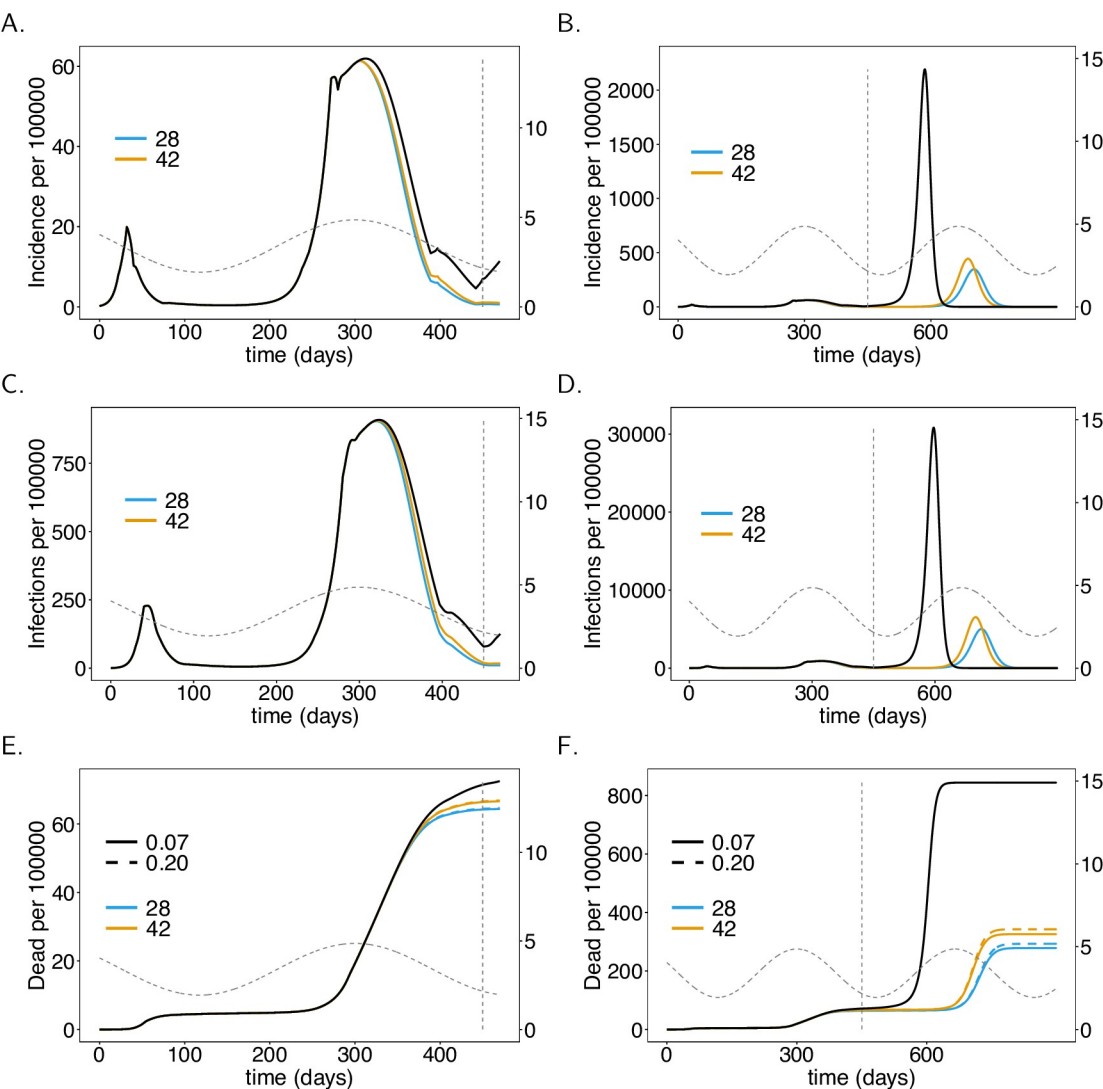

**Fig 7. Vaccination schedule and time to immune response.** As Fig 4, but for different average waiting times ($D_A$) until the vaccinations manifests its outcome (immunizing effect), corresponding to vaccination schedules and times to immune response.

## Fraction of the population being vaccinated

Also, the proportion of the population being vaccinated has a profound effect on disease incidence and mortality. Even if only 25% of the population gets vaccinated, there is a substantial decrease in incidence by time $t = 450$ (June 2021). The higher the proportion of individuals being vaccinated, the stronger the reduction in incidence (Fig 5A and 5C). Incidence will not substantially decrease further if the proportion of the population being vaccinated exceeds 80%. This directly translates into mortality (Fig 5E and 5F).

It takes a vaccination coverage of about 70% to avoid a high epidemic peak after the contact reductions are lifted. The lower the proportion of the population being vaccinated, the earlier and higher will the epidemic peak be (Fig 5B). If 75% of the population gets vaccinated, an epidemic peak emerges in spring 2022 ($t = 750$)—notice this prediction assumes that no contact reducing interventions are in place after summer 2021 ($t = 450$). This peak will exceed the one in early 2021. If 80% of the population is vaccinated, a pandemic peak that is in between the

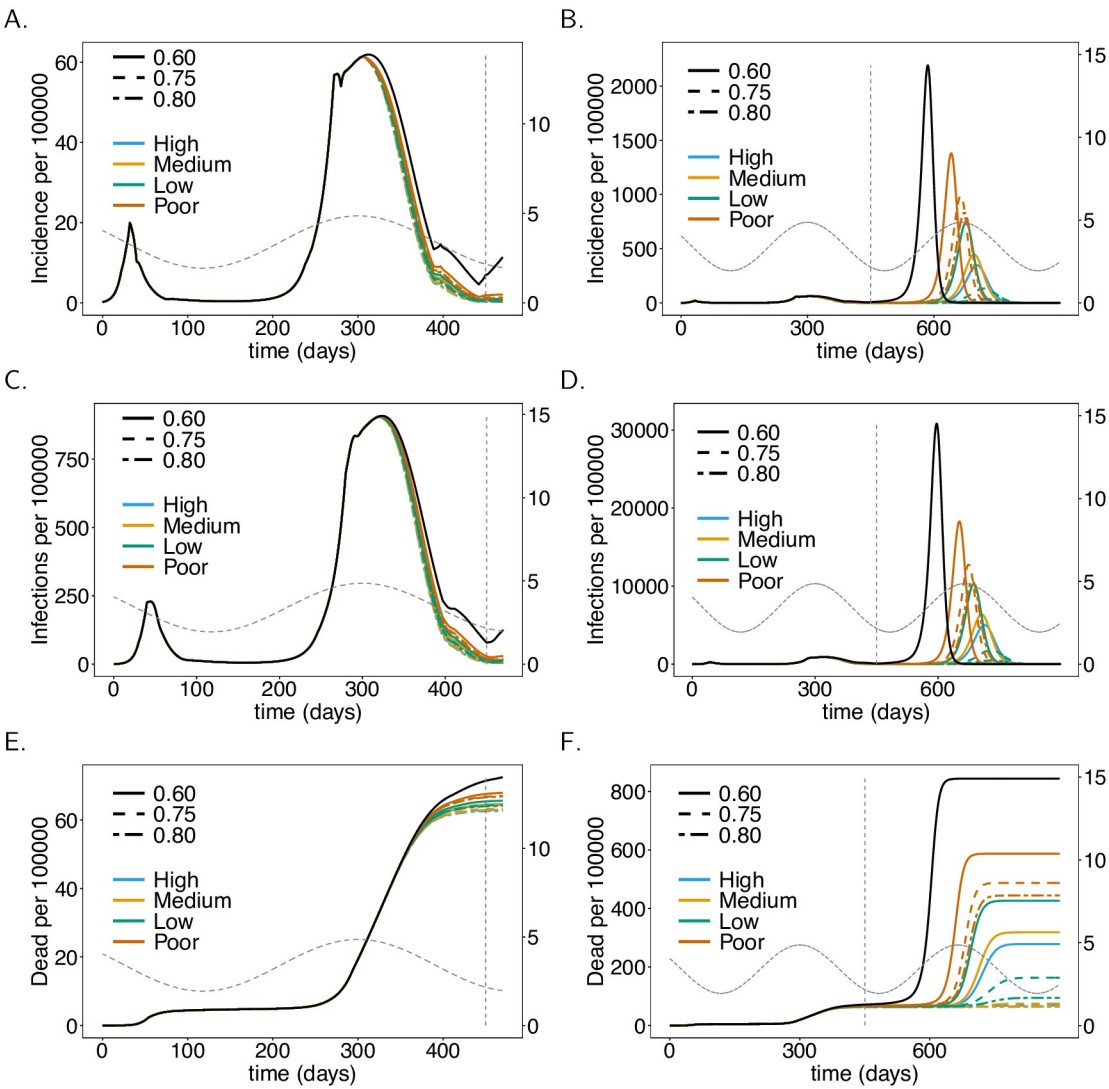

**Fig 8. Vaccine effectiveness.** As Fig 4, but for different vaccination coverage and effectiveness (quality) of the vaccine as summarized in Table 1. No ADE-induced increased mortality is assumed.

first and second wave of 2020 emerges. This peak will be avoided if 85% of the population gets vaccinated.

Mortality substantially decreases as the proportion of the population getting vaccinated increases (Fig 5E and 5F). The reduction occurs in a nonlinear fashion and shrinks with higher proportions of the population being vaccinated.

Note that vaccination coverage is not the same as the herd immunity threshold. The latter is the percentage of the population that needs to be immune for the disease to vanish. The vaccination coverage is the the percentage of the population that will be vaccinated throughout the epidemic. A higher coverage reflects faster deployment of the vaccine.

## Rate of vaccination

Not surprisingly, the faster the population is vaccinated the better. Assuming a realistic average waiting time to be vaccinated ranging between 180–300 days and 60% vaccination coverage,

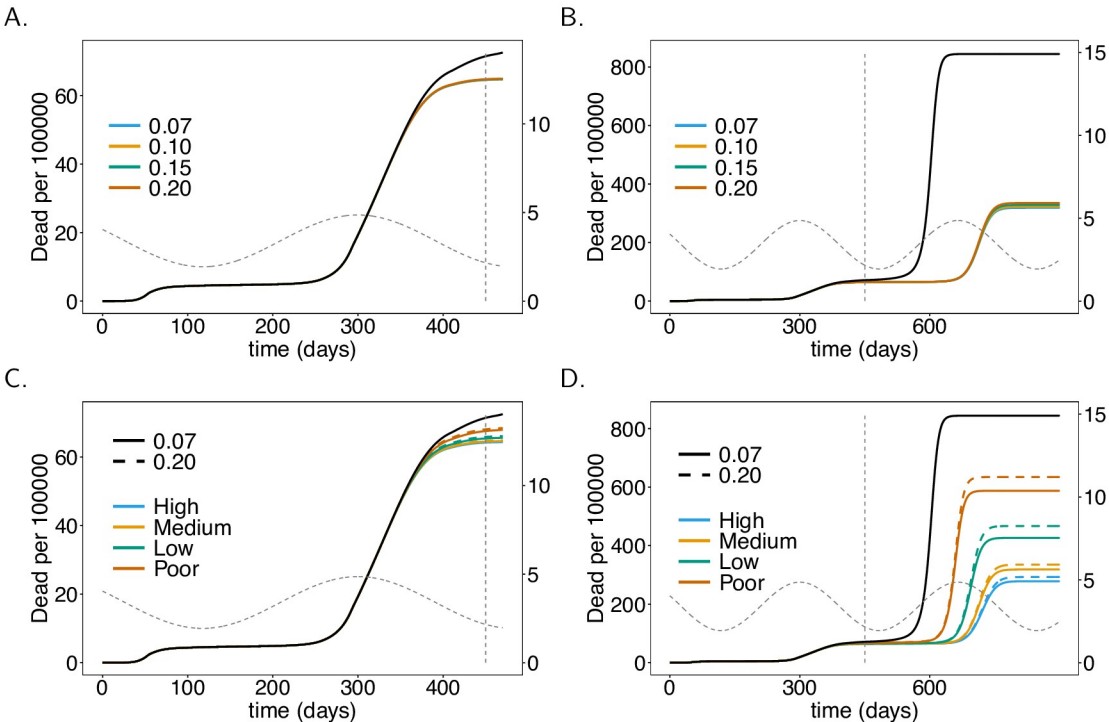

**Fig 9. ADE-induced increased mortality.** As Fig 8, but for different levels of ADE-induced mortality instead of vaccination coverage. A vaccination coverage of 60% was assumed. Panels show only mortality. For the corresponding incidence see Fig 8.

the effects are immediately visible in the short term (from $t = 300$ to $t = 450$, see Fig 6A and 6C). The benefit of vaccinating the population fast becomes substantial in the long term. In fact, the epidemic peak that emerges at around generation $t = 700$ (early 2022) is twice as high if the waiting time to get vaccinated is 300 days rather than 180 days on average (see Fig 6B and 6D). This translates directly into mortality (see Fig 6E and 6F).

## Vaccination schedule and immunogenicity

Another important factor in VCs is the time until immunization is reached, as determined by the vaccination schedule and immunogenicity. There is a visible effect if the time to immunization is increased from 28 days to 42 days. However, the effect is not as strong as that of the rate of infection or the proportion being vaccinated. In relative terms, the short-term effect is more pronounced (cf. Fig 7A–7D). The reason is that the number of infections is rising at the onset of the vaccination campaign. During this period early immunization reduces the spread of COVID-19. Once incidence is low, the time to immunization is not as important until the final epidemic peak emerges. This will emerge earlier and will be higher if the time to immunization is longer (see Fig 7B and 7D). Again, incidence directly translates into mortality (see Fig 7E and 7F). The long-term effect on mortality of longer times to immunization is by far less pronounced than those of the vaccination rate and proportion of vaccinated individuals.

## Effectiveness of the vaccine

The better the immunizing effect of the vaccine, the more effective is the VC. We compared four scenarios summarized in Table 1, one being a worst-case scenario which is unrealistic and just used as a comparison. When effectiveness increases from 78% (low) to 94% (high) full

**Table 1. Vaccine effectiveness.**

| Effectiveness | $f_.^{(R)}$ | $f_.^{(NI)}$ | $f_.^{(PI)}$ | $f_.^{(ADE)}$ |
|---|---|---|---|---|
| High | 0.94 | 0.02 | 0.03 | 0.01 |
| Medium | 0.90 | 0.04 | 0.05 | 0.01 |
| Low | 0.78 | 0.10 | 0.10 | 0.02 |
| Poor | 0.50 | 0.24 | 0.24 | 0.02 |

Parameter values corresponding to vaccine effectiveness, where the subscript "." is a placeholder for susceptibles ($S$), latent ($E$), prodromal ($P$) and fully infectious ($I$), respectively. After, the outcome of the vaccine manifests, $f_.^{(R)}$, $f_.^{(PI)}$, $f_.^{(NI)}$, $f_.^{(ADE)}$ are the fractions of individuals who become completely immune, partially immune, fail to immunize, and develop ADE, respectively.

immunization there will be a clear drop in disease incidence and mortality in the long run (see Fig 8B, 8D and 8F). The short-term effects are not as pronounced (see Fig 8A, 8C and 8E). The hypothetical worst-case scenario, in which the effectiveness is only 50%, leads to substantially higher infections and deaths.

## Severity of ADE

By default, we assumed that ADE manifests in a higher probability of developing symptomatic infections, i.e., $f_{\text{Sick}}^{(\text{ADE})} = 92\%$ vs. $f_{\text{Sick}} = 58\%$. Additionally, we investigated the effect of higher mortality of symptomatic infections in individuals that developed ADE. Figs 4–8 contrast the situations with and without ADE-induced increased mortality ($f_{\text{Dead}}^{(\text{ADE})} = 20\%$ vs. $f_{\text{Dead}} = f_{\text{Dead}}^{(\text{ADE})} = 7\%$). This increased mortality does not change incidence. For a vaccine with high effectiveness, the impact of ADE-induced increased mortality is marginal. If effectiveness is low, the effect is visible. This is not surprising, because the occurrence of ADE and vaccine effectiveness are not independent, namely, ADE occurs only if the vaccine fails to immunize properly. Fig 9 shows the effect of different amounts of mortality induced by ADE for different vaccination coverage. The effects are not immediate and only visible in the long run. The lower the vaccination coverage, the higher the increase in mortality. This is not surprising: if a higher proportion of the population is immunized, it is less likely that individuals that developed ADE get infected. Thus, the deleterious effects of ADE cannot manifest.

## Discussion

With COVID-19 vaccines being approved swift and efficient action is required to immunize populations around the globe to further contain health and economic damages caused by the pandemic. Vaccination strategies need likely to be adjusted flexibly in response to the current status of the pandemic, feasibility of logistics, further vaccines becoming available, potential contraindications, side and long-term effects that were not recognized during ongoing phase III clinical studies, and the willingness in the population to get vaccinated. The purpose of this study was to tailor predictive modelling to optimize vaccination strategies for COVID-19 management and eradication. More precisely, we developed a realistic pandemic-preparedness model to study the influence of the onset of vaccination campaigns, the vaccination rate (i.e., the average time to get vaccinated), the vaccination coverage (the fraction of the population that can get vaccinated), vaccination schedules, effectiveness of the vaccine, and adverse side effects, particularly ADE. The model is a complex extension of the model underlying CovidSim 1.1 http://covidsim.eu/ [18]. We parameterized the model to reflect the situation in the Federal

Republic of Germany, where vaccination campaigns started in late December 2020. Notably, although the results reported reflect the situation in Germany, the results apply qualitatively to any other country. Although, the model itself can be adapted to other countries to obtain quantitative results, their appropriateness needs to be taken with caution. Namely, the model neglects an explicit age-structure. Therefore, it is applicable to industrial nations with demographics similar to Germany. Adaptations will be necessary for low and middle-income countries with a large young population. These adaptations can be done similarly as in CovidSim 2.0 http://covidsim.eu/. To illustrate that the model is applicable to other countries (with similar demographic structure), we parameterized the models also to reflect the situation in the USA (see S2 Appendix).

The impact of the vaccination in terms of incidence and mortality depends on the contact-reducing interventions in place. Here, it was assumed that a "hard lockdown" will be sustained until the end of March 2020 followed by a "relief period" still with relatively strong contact reduction, until summer 2021. Afterwards no contact reduction was assumed. These assumptions are obviously questionable. The rationale behind them was that case numbers in 2021 will first require action to reduce disease incidence by sustaining contact reduction. Once people get vaccinated and incidence is decreasing, compliance with distancing measures will fade after the summer season. As soon as attendance of cultural events (e.g., concerts, museums, sports events, etc.) and unrestricted air travel will be possible and mandatory wearing of facial masks will be lifted for vaccinated individuals, it will be difficult to control distancing interventions in the population. Notably, we assume that case isolation (quarantine and home isolation of confirmed cases) is further sustained.

Our simulations adequately reflect the dynamics of the COVID-19 epidemic in Germany in 2020. The sensitivity on parameters not related to vaccination, can be ascertained from the CovidSim 1.1 web-simulator. The parameters used here were intuitively chosen and have a clear interpretation. It is very unlikely to obtain identical dynamics by choosing a totally different set of parameters, which still has plausible interpretations. The future predictions, however, depend on assumptions regarding COVID-19 management. The model parameters need to be dynamically updated as COVID-19 management interventions are altered. Particularly, parameters concerning vaccination campaigns are likely to vary over time and can be modelled in a time-dependent fashion. This is a straightforward generalization. Here, we decided to simulate a plausible range of fixed parameters to quantify their effects, rather than making assumptions on time-dependence that would be purely speculative. According to our predictions, vaccination campaigns will have a strong impact on the reduction in disease incidence. In the short term, a swift onset of the vaccination campaign contributes to a substantial reduction in incidence and mortality in the first quarter of 2021. The later mass vaccination starts, the smaller the reduction in incidence or mortality. The onset of the campaign mainly depends on the logistics to initially distribute the vaccine efficiently.

Importantly we assumed the optimistic case that the vaccine protects from infection and transmission. This however, is not clear yet. Vaccines might just protect from severe disease. In our model this situation can be accommodated, by assuming that the vaccine leads with a high probability only to partial immunity, which results in symptomatic infections with significantly reduced probability. However, data from Israel suggests that the BioNTech vaccine protects from transmission [21]. Moreover, we did not assume the British or South African mutation. Not all vaccines might protect from these variants [22–24]. Also these situations can be accommodated by the model. These mutations are characterized by a higher base reproductive number. Thus they will spread, which can be captured in our model, by increasing the yearly average base reproductive number in a sigmoidal fashion and adjust the parameters reflecting the effects of the vaccine in a similar fashion.

The vaccination schedule also has a substantial impact on incidence and mortality. Our simulations showed a substantial improvement if the time to immunization is shortened from 42 to 28 days. This depends on the vaccination schedule, which requires typically two doses with a waiting time of two to three weeks between them. After that, the immunizing effect is reached within about 14 days. Efficient logistic planning and properly scheduling appointments to receive the two required doses can help to minimize the time to immunization.

Not surprisingly, the higher the vaccination rate (i.e., the faster the population is vaccinated), the stronger the benefit. The vaccination rate depends crucially on the available infrastructure. The modRNA-based vaccine of BioNTech (cf. Introduction) requires storage between −60 to −80˚C [25]. Therefore, it needs to be distributed through a specific infrastructure (vaccination centers/units) limiting the vaccination rate. The achievable vaccination rate hence depends on the approval and availability of competing vaccines. Vaccine efficiency differs across competing products and has a strong effect on incidence and mortality. In the simulations, by default, we assumed an ambitious vaccination rate, with an average waiting time of 180 days to get vaccinated. With two doses per person vaccinating 60% of the German population within one year requires a capacity of 270000 injections per day. With around 400 vaccination centers that have been established, this requires a daily average capacity of 680 injections per center. While the capacity is realistic, also the willingness to get vaccinated must be high to efficiently utilize the capacities.

Another important factor is vaccination coverage. Our simulations suggest that a vaccination coverage of 75%−80% is necessary to mitigate the epidemic by summer 2021, without further strong lockdowns and contact restrictions. The reason is that a sufficiently high level of immunity needs to be reached by the onset of the 2021 flu season in order to prevent another epidemic outbreak. Such an outbreak might be more difficult to control as interventions will be a balancing act between restrictions tolerable by the vaccinated part and necessary to protect the unvaccinated part of the population. Notably, even if the herd immunity threshold to prevent a COVID-19 outbreak is substantially lower than 75%, this threshold must be reached on time. Importantly, the vaccination coverage in our predictions also reflects the propensity to get vaccinated early. Furthermore, vaccination does not lead to immunization in all cases, a fact that was addressed in our model. In particular, we studied the consequences of varying vaccine efficiencies, which have a substantial effect. According to clinical trials, efficacy varies between 78%−95% among the most promising vaccines [26–29]. With lower efficiency vaccination coverage must increase to reach herd immunity, i.e, to reach immunization in 60% if the population 63% needs to be vaccinated if efficiency is 95%, while 77% need to be vaccinated if efficiency is just 78%.

Our predictions are conservative in as far as negative side effects of the vaccine were considered. In particular, we incorporated ADE or more generally ERD. These are well known in corona viruses and it has been explicitly warned about ADE in the context of vaccination campaigns [17]. The effects of ADE are notoriously difficult to predict [30]. Here, we assumed that a fraction of vaccinated individuals develops ADE, which results in a higher likelihood to develop symptomatic infections and higher mortality. Although we assumed mortality to be substantially increased (20% rather than 7% mortality in symptomatic infections), the overall effect was minor. More precisely, the reduction in mortality due to immunization achieved through vaccination always outweighs increased mortality due to ADE. This is an encouraging result that justifies neglecting ADE in future models. In fact, our model can be substantially simplified if ADE is neglected.

While our predictions adequately reflect disease incidence, the predictions for mortality have to be interpreted with caution. Namely, vaccination campaigns will target risk groups suffering from elevated mortality first. In fact, 50% of COVID-19-related deaths occur in long-term care facilities (LTCFs), although less than 1% of the German populations live inside such

a facility. Under thorough contact reducing measures, the spread of COVID-19 inside LTCFs can be efficiently maintained by regularly testing employees [31]. Concerning the interpretations of our results, mortality has to be understood qualitatively rather than quantitatively. However, adequate quantitative predictions can be easily deduced from our simulations by multiplying mortality with an adjustment factor. The relative effect of model parameters remains unaffected by an increase or decrease in mortality.

Our results for the USA (see S2 Appendix) are similar than for Germany, although we predict that a lower vaccination coverage is sufficient to avoid a further epidemic peak. Notably, these results also do not assume the more infectious British mutation. However, they serve as a benchmark for comparison.

In conclusion, vaccination campaigns should be launched as early as possible. Logistics should be well planned to utilize the maximum capacity of the vaccination infrastructure. Failure to immunize a sufficient part of the population by the beginning of the flu season in 2021 will result in a high endemic peak, by far exceeding current levels of incidence. Adverse effects of the vaccine such as ADE are by far outweighed by the benefits of the vaccine. In fact, the higher vaccination coverage, the lower the risks associated with ADE. We predict that vaccination coverage of 80% would result in sufficiently high levels of herd immunity to allow a return to normality by summer 2021. Nevertheless, it is important to sustain the vaccination campaign until the herd immunity threshold is actually reached. This will require sustained incentives to get vaccinated after disease incidence drops, e.g., through general contact reductions measures being tight to vaccination coverage.

## Supporting information

**S1 Text.**
(TXT)

**S1 Fig. Model flow chart.** Illustration of the full model flow chart, showing all compartments (boxes) and possible transitions from compartments (arrows).
(PDF)

**S2 Fig. Onset of the vaccination campaign: As in Fig 4 but for U.S. instead of Germany.** Parameters for contact reduction are given in S7 Table.
(PDF)

**S3 Fig. Vaccination coverage.** As Fig 5, but for the USA instead of Germany.
(PDF)

**S4 Fig. Vaccination rate.** As Fig 6, but for the USA instead of Germany.
(PDF)

**S5 Fig. Vaccination schedule and time to immune response.** As Fig 7, but for the USA instead of Germany.
(PDF)

**S6 Fig. Vaccine effectiveness.** As Fig 8, but for the USA instead of Germany.
(PDF)

**S7 Fig. ADE-induced increased mortality.** As Fig 9, but for the USA instead of Germany.
(PDF)

**S1 Appendix. Mathematical description.**
(PDF)

**S2 Appendix. Results for the USA.**
(PDF)

**S1 Table. (Sub-) population sizes of Germany (GER) and the USA chosen in simulations.**
(PDF)

**S2 Table. Parameters describing disease progression for Germany (GER) and the USA.**
(PDF)

**S3 Table. Summary of variables describing sub-population sizes in Germany (GER) and the USA.**
(PDF)

**S4 Table. Parameters describing disease severity and mortality for Germany (GER) and the USA.**
(PDF)

**S5 Table. Parameters describing contact behavior and force of infection for Germany (GER) and the USA.**
(PDF)

**S6 Table. Contact reduction parameters chosen for the simulations of Germany.**
(PDF)

**S7 Table. Contact reduction parameters chosen for the simulations of the USA.**
(PDF)

## Acknowledgments

We want to dedicate this work to all voluntary participants in the COVID-19 vaccination trials. We also want to dedicate it to the victims of the SARS-CoV-2 virus. Our grief is with the friends and families of the dreadful disease. The authors gratefully acknowledge the discussion with Prof. Martin Eichner and Prof. Gideon Ngwa. The authors also want to express their gratitude to Dr. Jing Yuan, Dr. Preetida J Bhetariya and an anonymous reviewer for constructive comments that helped improving the manuscript.

## Author Contributions

**Conceptualization:** Nessma Adil Mahmoud Yousif, Henri Christian Junior Tsoungui Obama, Yvan Jordan Ngucho Mbeutchou, Sandy Frank Kwamou Ngaha, Loyce Kayanula, George Kamanga, Looli Alawam Nemer, Kristina Barbara Helle, Miranda Ijang Teboh-Ewungkem, Kristan Alexander Schneider.

**Data curation:** Nessma Adil Mahmoud Yousif, Henri Christian Junior Tsoungui Obama, Yvan Jordan Ngucho Mbeutchou.

**Formal analysis:** Nessma Adil Mahmoud Yousif, Henri Christian Junior Tsoungui Obama, Yvan Jordan Ngucho Mbeutchou, Sandy Frank Kwamou Ngaha, Loyce Kayanula, George Kamanga, Kristina Barbara Helle, Kristan Alexander Schneider.

**Funding acquisition:** Kristan Alexander Schneider.

**Investigation:** Nessma Adil Mahmoud Yousif, Henri Christian Junior Tsoungui Obama, Yvan Jordan Ngucho Mbeutchou, Sandy Frank Kwamou Ngaha, Loyce Kayanula, George

Kamanga, Toheeb Babatunde Ibrahim, Patience Bwanu Iliya, Looli Alawam Nemer, Kristina Barbara Helle, Kristan Alexander Schneider.

**Methodology:** Nessma Adil Mahmoud Yousif, Henri Christian Junior Tsoungui Obama, Yvan Jordan Ngucho Mbeutchou, Sandy Frank Kwamou Ngaha, Loyce Kayanula, George Kamanga, Toheeb Babatunde Ibrahim, Kristina Barbara Helle, Kristan Alexander Schneider.

**Project administration:** Nessma Adil Mahmoud Yousif, Kristan Alexander Schneider.

**Resources:** Kristan Alexander Schneider.

**Software:** Nessma Adil Mahmoud Yousif, Henri Christian Junior Tsoungui Obama, Yvan Jordan Ngucho Mbeutchou, Kristina Barbara Helle, Kristan Alexander Schneider.

**Supervision:** Kristan Alexander Schneider.

**Validation:** Nessma Adil Mahmoud Yousif, Henri Christian Junior Tsoungui Obama, Kristan Alexander Schneider.

**Visualization:** Nessma Adil Mahmoud Yousif, Henri Christian Junior Tsoungui Obama, George Kamanga, Kristan Alexander Schneider.

**Writing – original draft:** Nessma Adil Mahmoud Yousif, Henri Christian Junior Tsoungui Obama, Yvan Jordan Ngucho Mbeutchou, Sandy Frank Kwamou Ngaha, Loyce Kayanula, George Kamanga, Toheeb Babatunde Ibrahim, Patience Bwanu Iliya, Sulyman Iyanda, Looli Alawam Nemer, Kristina Barbara Helle, Miranda Ijang Teboh-Ewungkem, Kristan Alexander Schneider.

**Writing – review & editing:** Nessma Adil Mahmoud Yousif, Henri Christian Junior Tsoungui Obama, Yvan Jordan Ngucho Mbeutchou, Sandy Frank Kwamou Ngaha, Loyce Kayanula, George Kamanga, Toheeb Babatunde Ibrahim, Patience Bwanu Iliya, Sulyman Iyanda, Kristina Barbara Helle, Miranda Ijang Teboh-Ewungkem, Kristan Alexander Schneider.

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
