## [Decision Letter · Decision Letter 0]

4 Feb 2021

PONE-D-20-40700

The impact of COVID-19 vaccination campaigns accounting for antibody-dependent enhancement

PLOS ONE

Dear Dr. Schneider,

Thank you for submitting your manuscript to PLOS ONE. After careful consideration, we feel that it has merit but does not fully meet PLOS ONE’s publication criteria as it currently stands. Therefore, we invite you to submit a revised version of the manuscript that addresses the points raised during the review process.

Please review concerns raised by the reviewers and provide point by point response in your revised manuscript.

We look forward to receiving your revised manuscript.

Kind regards,

Muhammad Adrish

Academic Editor

PLOS ONE

Journal Requirements:

4. Please amend your authorship list in your manuscript file to include author Kristina Barbara Helle.

Reviewers' comments:

Reviewer's Responses to Questions

**Comments to the Author**

1. Is the manuscript technically sound, and do the data support the conclusions?

Reviewer #1: Yes

Reviewer #2: Yes

Reviewer #3: Yes

2. Has the statistical analysis been performed appropriately and rigorously? 

Reviewer #1: N/A

Reviewer #2: N/A

Reviewer #3: Yes

3. Have the authors made all data underlying the findings in their manuscript fully available?

Reviewer #1: Yes

Reviewer #2: Yes

Reviewer #3: Yes

4. Is the manuscript presented in an intelligible fashion and written in standard English?

Reviewer #1: Yes

Reviewer #2: Yes

Reviewer #3: Yes

5. Review Comments to the Author

Reviewer #1: Vaccination is an important way to protect people against covid-19 and will help the world return to normality. In this study authors employed a complicated SEIR model to explore the impact of vaccination campaigns. A very detailed realisation of the infection phases and immunization effects (both lethal and beneficial) of vaccination is good and bad at the same time. It is good because it closely describes the realities; it is bad because it introduces too many model parameters most of which may be hard to estimate. With values of many model parameters unknown or having large variations, model predictions are problematic. Although authors argued that the values of model parameters they used "reflect the situation in the Federal Republic of Germany. ", "The simulation results until late December (t = 300) match the disease incidence in Germany", it is hard to know how they obtained those values. In principle, it is possible to set another set of values which can also "match the disease incidence in Germany". To justify those values, one suggestion is to estimate the relevant model parameters using the data until December 2020.

In modelling of infectious diseases, sensitivity analysis is important to guarantee the prediction results. That is, how robust the model conclusions are under different combinations of model parameters.

Another issue is about the cost-effective. In Introduction, authors mentioned many productions of covid-19 vaccine and their costs. In view of cost-effectiveness, what is the best vaccination campaign?

Reviewer #2: I find the manuscript presents a predictive model with high confidence specifically designed for mid-size population countries to prepare for COVID19 vaccination strategies. Such tools are of great importance to public health administrations so that they can plan strategies such as prepare the healthcare workers for more rigorous countrywide vaccine campaigns and also prepare for the availability of enough vaccine doses. The manuscript is very well written and only needs few corrections. Some of the comments are attached.

Reviewer #3: The article is well written. The author used the extended model of the SEIR to predict the occurrence of ADE during vaccination campaigns in the ongoing COVID-19 pandemic. Several different vaccination schedules were assumed and their effects were analyzed using the extended SEIR model, including the influence of the onset of vaccination campaigns, the vaccination rate, the vaccination coverage, vaccination schedules, effectiveness of the vaccine, and adverse side effects and so on. Also, the model was parameterized to reflect the situation in the Federal Republic of Germany that so far intervened successfully in the COVID-19 pandemic. The results matched the situation in Germany. But the applicability of the model to other countries needs to be considered. To some extent, this model can predict the effect of the vaccination campaign.

Minor comments:

1. “invective” should be “infective” instead in Fig 1 legend.

2. “an” should revised as “and” on line 178.

6. PLOS authors have the option to publish the peer review history of their article (what does this mean?). If published, this will include your full peer review and any attached files.

Reviewer #1: No

Reviewer #2: **Yes: **Preetida J Bhetariya

Reviewer #3: **Yes: **Jing Yuan

---

## [Decision Letter · Decision Letter 1]

30 Mar 2021

The impact of COVID-19 vaccination campaigns accounting for antibody-dependent enhancement

PONE-D-20-40700R1

Dear Dr. Schneider,

We’re pleased to inform you that your manuscript has been judged scientifically suitable for publication and will be formally accepted for publication once it meets all outstanding technical requirements.

Kind regards,

Muhammad Adrish, MD, MBA, FCCP, FCCM

Academic Editor

PLOS ONE

Additional Editor Comments (optional):

All comments and suggestions have been addressed.

Reviewers' comments:

Reviewer's Responses to Questions

**Comments to the Author**

1. If the authors have adequately addressed your comments raised in a previous round of review and you feel that this manuscript is now acceptable for publication, you may indicate that here to bypass the “Comments to the Author” section, enter your conflict of interest statement in the “Confidential to Editor” section, and submit your "Accept" recommendation.

Reviewer #1: All comments have been addressed

Reviewer #3: All comments have been addressed

2. Is the manuscript technically sound, and do the data support the conclusions?

Reviewer #1: Yes

Reviewer #3: Yes

3. Has the statistical analysis been performed appropriately and rigorously? 

Reviewer #1: N/A

Reviewer #3: Yes

4. Have the authors made all data underlying the findings in their manuscript fully available?

Reviewer #1: Yes

Reviewer #3: Yes

5. Is the manuscript presented in an intelligible fashion and written in standard English?

Reviewer #1: Yes

Reviewer #3: Yes

6. Review Comments to the Author

Reviewer #1: (No Response)

Reviewer #3: The revised article well solves the problems I raised, so I think the article can be published. I have no comment to the author.

7. PLOS authors have the option to publish the peer review history of their article (what does this mean?). If published, this will include your full peer review and any attached files.

Reviewer #1: No

Reviewer #3: No

---

## [Editor Report · Acceptance letter]

12 Apr 2021

PONE-D-20-40700R1

The impact of COVID-19 vaccination campaigns accounting for antibody-dependent enhancement 

Dear Dr. Schneider:

I'm pleased to inform you that your manuscript has been deemed suitable for publication in PLOS ONE. Congratulations! Your manuscript is now with our production department.

Kind regards,

on behalf of

Dr. Muhammad Adrish

Academic Editor

PLOS ONE